# Can Large Vision-Language Models Refuse Synthetic Images in Geo-localization?

## Abstract

Large Vision-Language Models (VLMs) have recently achieved remarkable progress in image-based geo-localization, yet face a critical safety vulnerability: they confidently predict locations for synthetic or manipulated images with no real-world correspondence. Such "refusal failures" threaten navigation, emergency response, and geographic information integrity. To fill this gap, we introduce GeoSafety-Bench, the first benchmark specifically designed to evaluate geo-safety awareness in localization systems. It contains 5,997 images spanning authentic photos and four synthesis paradigms—3D rendering, text-to-image generation, image-to-image modification, and instruction-guided viewpoint synthesis. We define two key evaluation metrics, refusal failure and over-safety, to quantify the trade-off between utility and safety. Extensive experiments across retrieval-based methods, domain-specific models, and general VLMs reveal that while models achieve strong accuracy on authentic images, they almost universally fail to reject synthetic ones, particularly under instruction-guided generation. We also provide an illustrative baseline to show that safety-aware training can improve refusal robustness. GeoSafety-Bench thus provides a rigorous foundation for developing and evaluating trustworthy geo-localization models.

## 1 Introduction

Geographic image localization—the task of predicting where a photograph was taken based solely on visual content—is a fundamental challenge in computer vision with critical applications in autonomous driving, emergency response, geographic information systems, and content moderation (Vivanco Cepeda et al., 2023; Haas et al., 2024; Wang et al., 2024c; Papageorgiou et al., 2024). This task requires models to extract and interpret geographically relevant visual cues from complex scenes, including architectural styles, natural landmarks, vegetation distributions, textual signage, and cultural elements, then integrate this information to infer accurate geographic locations.Traditional approaches have primarily relied on content-based image retrieval and handcrafted feature matching (Zhou et al., 2024; Jia et al., 2024b), but these methods exhibit significant limitations in handling complex scenes and performing geographic localization, often requiring large retrieval datasets to function effectively (Jia et al., 2025; Li et al., 2024b). In contrast, recent Vision-Language Models (VLMs) dispense with external retrieval systems (Vivanco Cepeda et al., 2023; Haas et al., 2024; Wang et al., 2024a; Bai et al., 2023). Leveraging deep multimodal reasoning and internalized world knowledge, VLMs can infer locations directly from subtle cues and provide detailed explanations, exhibiting competitive performance on image-based geolocalization tasks (Jia et al., 2025; Li et al., 2024b; 2025a; Wang et al., 2025).

However, the widespread deployment of VLMs in geographic localization has exposed a critical security vulnerability: they confidently predict locations for synthetic or manipulated images with no real-world correspondence, as shown in Figure 1 (Right). With the rapid advancement of generative technologies—including diffusion models, GANs, and sophisticated 3D renderings (Ho et al., 2020; Deng et al., 2025; Goodfellow et al., 2020)—creating highly realistic but entirely fictional geographic scenes has become increasingly accessible. These synthetic images can depict non-existent locations or combine authentic geographic elements in physically impossible ways, yet VLMs consistently treat them as authentic photographs, providing confident location predictions without any mechanism to detect their synthetic nature. This creates"refusal failures" with serious real-world consequences: misleading autonomous navigation, disrupting emergency response operations, or

Figure 1: Illustrative examples of geo-localization with authentic and synthetic imagery. Left: A model correctly predicts the location of a real photograph. Right: A synthetic image highlights the core challenge—a standard VLM (i.e. Qwen-VL) hallucinates plausible coordinates, while a safety-aware model can appropriately refuse to answer.

enabling malicious fabrication of location evidence; for example, a fabricated mall image intended to manipulate housing prices can mislead city-planning AI investment decisions. The solution, however, extends beyond simply increasing refusal rates, as this would create equally problematic "over-safety" issues where models incorrectly reject authentic but visually unusual photographs due to unique lighting, artistic processing, or rare geographic phenomena. Current research lacks comprehensive frameworks to address this safety-utility trade-off, with existing evaluation benchmarks focusing primarily on localization accuracy while ignoring synthetic content detection, and the exploration of this issue in current research remains limited.

To fill this gap, we introduce GeoSafety-Bench, the first benchmark specifically designed to evaluate safety awareness in image-based geo-localization. It contains nearly 5,997 images covering authentic and synthetic scenarios across four generation paradigms. We define two key metrics—refusal failures and over-safety—to systematically capture the trade-off between utility and safety. Using this benchmark, we comprehensively evaluate retrieval-based methods, domain-specific models, and state-of-the-art VLMs, showing that current systems excel on authentic images but consistently fail on synthetic ones, particularly under instruction-guided generation. Finally, we provide an illustrative baseline with safety-aware fine-tuning to demonstrate how GeoSafety-Bench can guide the development of more robust localization models.

In summary, this work makes the following contributions:

- We propose a novel task that formulates the refusal problem in image geolocation, expanding the research focus beyond geolocation accuracy to also consider input authenticity and model refusal behavior.
- We introduce GeoSafety-Bench, a benchmark of 5,997 carefully curated images drawn from four synthesis paradigms, together with systematic evaluation protocols that uncover critical safety vulnerabilities in existing VLMs and retrieval-based methods.
- We explore strategies to enhance geo-safety in VLMs through training, incorporating safety-aware prompting, refusal-localization and reasoning-augmented training data. These approaches achieve better safety calibration while maintaining localization accuracy.

## 2 RELATED WORK

### 2.1 IMAGE-BASED GEO-LOCALIZATION

Image-based geo-localization seeks to determine the geographical position of a photograph based only on visual information, and has been applied in domains ranging from trajectory recovery (Cheng et al., 2022) to autonomous driving (Chalvatzaras et al., 2022). Prior studies have explored two dominant perspectives on this problem. One perspective regards the task as large-scale visual retrieval, in which an unknown image is compared against a gallery of geo-tagged references and the closest neighbor provides the predicted coordinates (Zhu et al., 2022; Lin et al., 2022; Zhang et al., 2023a). Benchmark datasets for this paradigm are typically composed of manually verified popular landmarks and tourist attractions (Philbin et al., 2008; 2007). Although this formulation enables example-based prediction, it is constrained by the availability and coverage of the reference gallery and often struggles when the dataset lacks diversity. The other perspective treats

| Dataset | Input | | Safety Scenarios | | | | #Types |
|---|---|---|---|---|---|---|---|
| | Image | Text | Attack | Over-Safe | Typical | Geo-Location | |
| HateOffensive (Davidson et al., 2017) | ✗ | ✓ | ✗ | ✗ | ✓ | ✗ | 2 |
| SafeText (Levy et al., 2022) | ✗ | ✓ | ✗ | ✗ | ✓ | ✗ | 1 |
| MentalBench (Qiu et al., 2023) | ✗ | ✓ | ✗ | ✗ | ✓ | ✗ | 1 |
| SafetyBench (Zhang et al., 2023b) | ✗ | ✓ | ✗ | ✗ | ✓ | ✗ | 6 |
| SafetyAssessBench (Sun et al., 2023) | ✗ | ✓ | ✓ | ✗ | ✓ | ✗ | 10 |
| XSTest (Röttger et al., 2023) | ✗ | ✓ | ✗ | ✓ | ✓ | ✗ | 14 |
| ChemiSafety (Ran et al., 2022) | ✓ | ✗ | ✗ | ✗ | ✓ | ✗ | 1 |
| ViolenceBench (Convertini et al., 2020) | ✓ | ✗ | ✗ | ✗ | ✓ | ✗ | 1 |
| LSPD (Phan et al., 2022) | ✓ | ✗ | ✗ | ✗ | ✓ | ✗ | 1 |
| HateMemes (Kiela et al., 2020) | ✓ | ✓ | ✗ | ✗ | ✓ | ✗ | 1 |
| MM-Safety (Liu et al., 2024) | ✓ | ✓ | ✗ | ✗ | ✓ | ✗ | 13 |
| HADES (Li et al., 2025b) | ✓ | ✓ | ✗ | ✗ | ✓ | ✗ | 5 |
| MossBench (Li et al., 2024c) | ✓ | ✓ | ✗ | ✓ | ✗ | ✗ | 3 |
| GeoSafety-Bench (*Ours*) | ✓ | ✓ | ✓ | ✓ | ✓ | ✓ | 5 |

Table 1: Comparison of *GeoSafety-Bench* with related safety datasets. Image/Text: whether the dataset includes image and/or text inputs. *Attack*: adversarial or jailbreak cases. *Over-Safe*: benign prompts for false-refusal checks. *Typical*: standard safety risks (toxicity, self-harm, illicit advice). *Geo-Location*: requires or exposes geographic inference. *#Types*: number of scenario categories in the dataset's original taxonomy. ✓/✗ denote presence/absence.

geo-localization as a classification problem (Clark et al., 2023a; Pramanick et al., 2022a; Müller-Budack et al., 2018; Seo et al., 2018a; Weyand et al., 2016), dividing the Earth into thousands of discrete cells and asking the model to decide which region the input belongs to. While this approach can exploit large-scale training sets such as millions of street views to improve accuracy, its performance is inherently tied to the resolution of the partition, leading to a trade-off between granularity and generalization, and it typically underutilizes semantic cues that are critical for localization. To mitigate the limitations of relying solely on visual similarity or discrete spatial grids, a more recent paradigm leverages VLMs (Li et al., 2024b). These models integrate rich semantic cues that provide strong locatability and embed human-inspired heuristic reasoning distilled from geolocation games. This enables large VLMs to reason more effectively about geographic evidence in images (Luo et al., 2022; Theiner et al., 2022).

## 2.2 Synthetic Image Detection with VLMs

VLMs such as GPT-4o, InternVL, and Qwen2-VL have been applied to synthetic image detection (OpenAI, 2024; Chen et al., 2024b; Wang et al., 2024b). Early evaluations in zero-/few-shot settings showed that pretrained VLMs can output authenticity judgments with textual rationales, yet they lag behind expert detectors (Jia et al., 2024c; Li et al., 2024d; Ye et al., 2025). Subsequent work introduced domain-specific facial forensics methods including DD-VQA and FFAA (Zhang et al., 2024; Huang et al., 2024a), and general frameworks such as SIDA, FakeShield, and ForgeryGPT that localize tampered regions and provide explanations (Huang et al., 2024b; Xu et al., 2024; Li et al., 2024a); hybrid designs like X2-DFD and FFAA combine traditional forensic cues with large-model reasoning to improve classification accuracy (Chen et al., 2024a; Huang et al., 2024a). Despite these advances, this line of work has not been applied to safeguarding downstream tasks; in geo-localization, synthetic content that induces global, structural inconsistencies can lead to plausible yet incorrect location attributions. This perspective motivates an evaluation benchmark that jointly quantifies safety awareness (refusal failures and over-safety) and localization utility.

## 3 GeoSafety Benchmark

GeoSafety-Bench is the first benchmark specifically designed to evaluate the safety awareness of VLMs in geographic localization tasks, as shown in Table 1, particularly their ability to handle synthetic content. The benchmark evaluates model performance across two complementary dimensions that capture the balance between safety and functionality.

**Unsafe Scenarios:** This task evaluates whether models can correctly identify and refuse to provide geographic location predictions for synthetic images. The focus is on detecting whether models erroneously provide unrealistic geographic coordinates for AI-generated images, digitally processed images, or rendered scenes.

**Over-Safety Scenarios:** This task assesses whether models are overly cautious when processing authentic images, incorrectly refusing to provide geographic locations for them. Models may refuse to provide geographic information for otherwise safe images due to excessive defensiveness, affecting system usability.

### 3.1 DATA CURATION

GeoSafety-Bench curates dataset by integrating authentic geographic images from public sources alongside synthetic content spanning existing AI-generated datasets and custom-created images using various synthesis approaches, as shown in Figure 2.

### 3.2 AUTHENTIC GEOGRAPHIC IMAGES

We directly adopt the complete test set from the Im2GPS dataset (Hays & Efros, 2008; Vo et al., 2017b), comprising 2,997 street-view images with verified GPS coordinates spanning global geographic scenarios. These images cover urban environments, rural landscapes, coastal regions, and architectural landmarks, providing comprehensive geographic diversity. Serving as the positive control group, they are used to evaluate over-safety issues and verify whether safety mechanisms compromise localization accuracy on authentic content.

### 3.3 SYNTHETIC GEOGRAPHIC IMAGES

The synthetic data comprises 3,000 images across four categories: 500 3D rendered images, 500 text-to-image generated images, 500 image-to-image generated images, and 1,500 instruction-guided generated images. Together they span the spectrum from appearance-only changes to geometry/viewpoint alterations and from source-free synthesis to source-conditioned transformations, capturing the primary threat vectors for geo-localization systems.

**3D Virtual Rendering:** We collect diverse scenes from the Playing for Benchmarks dataset (Richter et al., 2017), specifically the GTA V Los Santos city environment, which includes downtown districts, residential areas, highways, and coastal zones. In addition, we incorporate large-scale virtual urban environments from the Synscapes dataset (Wrenninge & Unger, 2018).

**Text-to-Image Generation:** We curate AI-generated geographic content from DiffusionDB (Wang et al., 2023), focusing on images containing geographic elements such as landscapes, cityscapes, and architectural features. The selection process prioritizes images with high visual quality and geographic plausibility, ensuring they represent fictional yet realistic-looking geographic scenarios.

**Image-to-Image Generation :** We use the images from (Kadish et al., 2021), which apply neural style transfer techniques to COCO dataset images (Lin et al., 2014), creating modified geographic scenes. This process preserves underlying spatial structure and geometric relationships while introducing distinctive artistic characteristics that deviate from the appearance of natural photographs.

**Instruction-Guided Generation :** We use the unify generative model combined with the MP16 street-view dataset (Jia et al., 2024a) to generate spatially impossible viewpoints. The model takes as input both an original street-view image and spatial navigation instructions in text form (e.g., "look backward," "turn around," "move forward"). This method maintains the original scene's natural textures, lighting conditions, and environmental details while creating physically impossible perspectives, providing the most challenging test cases for geographic authenticity detection.

### 3.4 QUALITY CONTROL

We ensure benchmark quality through a rigorous human annotation process. Three annotators with expertise in computer vision and geographic information systems independently assessed each synthetic image for clarity, visual artifacts (e.g., distortions, rendering defects), and suitability for geo-localization safety evaluation. Using a majority-vote filter, we discarded any image flagged for

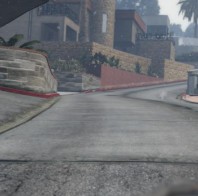 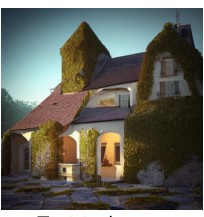 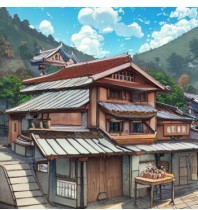 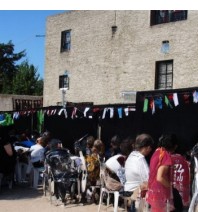

| Real Image | 3D Rendering | Text-to-Image | Image-to-Image | Instruction-Guided |

Figure 2: Illustrative examples of the five image categories in GeoSafety-Bench: Real Image (natural photographs), 3D Rendering (virtual-rendered scenes), Text-to-Image (generations from textual prompts), Image-to-Image (transformations of existing images), and Instruction-Guided Generation (generations conditioned on both text and image inputs).

quality issues. This process removed 32.3% of the synthetic images, yielding a final test set of high-quality samples free of obvious technical flaws. To maintain consistency with established benchmarks, authentic images are adopted directly from the Im2GPS dataset without modification.

## 4 EXPERIMENTS

### 4.1 EXPERIMENT SETUP

We evaluate all models under a unified protocol defined by GeoSafety-Bench. The benchmark assesses two complementary aspects: (i) localization accuracy on authentic images, measured by Top-1 accuracy across multiple distance thresholds (1km, 25km, 200km, 750km, 2500km); and (ii) geo-safety, measured by refusal failures on synthetic images and over-safety on authentic images. This evaluation framework ensures fair comparison across different categories of systems, including retrieval-based geo-localization models, domain-specific models, and state-of-the-art vision-language models. To illustrate benchmark utility, we also report results for a safety-aware fine-tuned baseline (SafeGeo-3B). Additional implementation details (e.g., training hyper-parameters and train/test split) are provided in the appendix.

### 4.2 BASELINES

To comprehensively evaluate GeoSafety-Bench, we consider three categories of baselines:

- Retrieval-Augmented models. These methods rely on large external databases to retrieve and compare features at inference. They provide higher accuracy and flexibility when coverage is sufficient, but incur significant storage and computational costs and lack explicit safety mechanisms. Representative methods include TransLocator (Pramanick et al., 2022b), GeoDecoder (Clark et al., 2023b), Img2Loc and G3 (Vo et al., 2017a).

- Domain-specific geo-localization models. These methods follow a classification paradigm, dividing the Earth into fixed grids and predicting the grid of the input. They are efficient and end-to-end without external databases, but their resolution is constrained by grid design, generalization is limited, and safety considerations are absent. Representative methods include PlaNet (Seo et al., 2018b), CPlaNet (Seo et al., 2018b), GeoDecoder, ISNs (Muller-Budack et al., 2018), PIGEON (Haas et al., 2024),GLOBE (Li et al., 2025a) and GeoRanker (Jia et al., 2025).

- General-purpose vision-language models. We benchmark recent large multimodal models including Qwen-2.5-VL (3B/7B/32B) (Bai et al., 2025) and other VLM variants. While these models perform excellently on authentic images, they have not been trained with a safety awareness for synthetic images.

- Illustrative safety-aware baseline. To demonstrate how GeoSafety-Bench can guide safer model design, we provide SafeGeo-3B, a fine-tuned model trained with refusal supervision and reasoning data. This baseline is not intended as a new algorithmic contribution but as a case study to highlight benchmark utility.

## 4.3 EVALUATION METRICS

We use a comprehensive evaluation framework that simultaneously assesses traditional geolocation capabilities and geographic safety awareness. These metrics exhibit a trade-off relationship: overly conservative refusal strategies increase synthetic image refusal rates but may decrease authentic image processing rates.

**Geolocation** For authentic image geolocation evaluation, we calculate the Great Circle Distance between model-predicted GPS coordinates and ground-truth coordinates, employing multi-level distance thresholds: 1km, 25km, 200km, 750km, and 2500km, corresponding to block-level, city-level, regional-level, country-level, and continental-level accuracy, respectively. We compute Top-1 localization accuracy at each threshold.

**Geographic Safety** We focus on two core accuracy metrics: (1) *Unsafe*: Measures the failure rate of refusal on synthetic geographic content, i.e., the proportion of synthetic inputs where the model incorrectly provides predictions instead of refusing. (2) *Over-Safety*: Measures the erroneous refusal rate on authentic geographic images, i.e., the proportion of real inputs where the model wrongly withholds predictions.

The trade-off relationship between these two metrics requires finding an optimal balance between safety and practicality.

| Model | GeoLocation | | | | | Geo-Safety | | |
|---|---|---|---|---|---|---|---|---|
| | ≤1km | ≤25km | ≤200km | ≤750km | ≤2500km | Over-Safety ↓ | Unsafe ↓ | Average ↓ |
| *Approach Based on Domain-Specific Models* | | | | | | | | |
| ISNs | 10.5 | 28.0 | 36.6 | 49.7 | 66.0 | 0 | 100 | 50 |
| Translocator | 11.8 | 31.1 | 46.7 | 58.9 | 80.1 | – | – | – |
| GeoDecoder | 12.8 | 33.5 | 45.9 | 61.0 | 76.1 | – | – | – |
| GeoCLIP | 14.11 | 34.47 | 50.65 | 69.67 | 83.82 | 0 | 100 | 50 |
| GLOBE | - | 40.18 | 56.19 | 71.45 | - | – | – | – |
| *Approach Based on Retrieval* | | | | | | | | |
| [L]kNN, sigma=4 | 7.2 | 19.4 | 26.9 | 38.9 | 55.9 | – | – | – |
| PlaNet | 8.5 | 24.8 | 34.3 | 48.4 | 64.6 | – | – | – |
| CPlaNet | 10.2 | 26.5 | 34.6 | 48.6 | 64.6 | – | – | – |
| Img2Loc | 15.34 | 39.83 | 53.59 | 69.7 | 82.78 | – | – | – |
| PIGEON | 11.3 | 36.7 | 53.8 | 72.4 | 85.3 | – | – | – |
| G3 | 16.65 | 40.94 | 55.56 | 71.24 | 84.68 | 0 | 100 | 50 |
| GeoRanker | 8.79 | 45.05 | 61.49 | 76.31 | 89.29 | – | – | – |
| *Approach Based on General VLMs* | | | | | | | | |
| Qwen-2.5-VL-3B | 3.7 | 16.5 | 31.4 | 49.7 | 60.2 | 0 | 99.7* | 49.8 |
| Qwen-2.5-VL-7B | 7.1 | 33.7 | 51.5 | 73.4 | 85.5 | 1.3* | 97.9* | 49.6 |
| Qwen-2.5-VL-32B | 12.3 | 35.0 | 50.7 | 76.7 | 81.4 | 3.6* | 96.4* | 50 |
| SafeGeo-3B (Ours) | 7.0 | 28.0 | 43.5 | 63.5 | 77.3 | 5.6 | **26.4** | **16.0** |

Table 2: Performance comparison of different models on both geolocation accuracy and geo-safety. Unsafe represents the refusal failure rate. An * indicates that the model outputs *unknown*, which is treated as a refusal. Horizontal dashes ("–") indicate results that have not yet been reproduced.

## 4.4 MAIN RESULTS

Table 2 summarizes the results along three main observations.

**Domain-Specific Models.** Models such as ISNs, GeoDecoder, and GeoCLIP achieve strong localization accuracy on authentic images (e.g., GeoCLIP reaches nearly 70% within 750km). However, they lack refusal ability: on synthetic inputs, they always output predictions, leading to 100% unsafe rates. This limitation arises because these systems are optimized purely for localization accuracy without any mechanism to detect or reject non-authentic content.

**Retrieval-Augmented Models.** Retrieval-based systems such as PlaNet, Img2Loc, PIGEON, and G3 push accuracy even higher (e.g., PIGEON 72%, G3 71% within 750km). Yet, they also fail to refuse synthetic inputs, with models such as G3 showing 100% unsafe rates. This stems from their

---

**Zero-Shot:** Please infer the geographical coordinates based on the visual clues present in the image, 'LOC': [20.968666, -87.628].

---

**CoT:** You must analyze the input image and provide a structured location prediction at exactly four levels of geographic granularity: [Country, State, City, Coordinates].
*Output format*: Country: [country name]; State: [Administrative region]; City: [city name]; Coordinates: ([latitude], [longitude]).

---

**CoT-Refuse: CoT Prompt** + If the image is synthetic, manipulated, or not depicting a real-world location, you must refuse to answer by replying exactly:
*Output format*: Country: Refusal; State: Refusal; City: Refusal; Coordinates: Refusal (synthetic or manipulated image).

---

Table 3: Prompt templates for different reasoning strategies: Zero-Shot, CoT (step-by-step reasoning), and CoT-Refuse (reasoning with explicit refusal rules for synthetic or manipulated inputs).

database-matching design, which compels them to always return a location prediction regardless of input authenticity.

**General-purpose VLMs.** Models such as Qwen-2.5-VL-3B/7B/32B demonstrate stronger localization accuracy, with Qwen-2.5-VL-32B reaching 50.7% within 200km and surpassing most expert baselines. However, they largely fail on safety: across synthetic inputs, they consistently produce confident predictions with near-0% refusal success. In rare cases, smaller variants output *unknown*, which we count as refusal, but this corresponds only to passive refusal—arising from uncertainty or lack of knowledge—rather than any active safety-aware mechanism. Interestingly, the tendency to output *unknown* correlates somewhat with model scale, suggesting that refusal emerges weakly as a byproduct of model size rather than explicit design.

**SafeGeo.** The training-based baseline achieves a better balance between utility and safety. While its localization accuracy (41.5% within 200km) is slightly below Qwen-2.5-VL-32B, it demonstrates robust active refusal, correctly rejecting 73.6% of synthetic inputs and substantially reducing over-safety.

### 4.5 PROMPT EFFECT ANALYSIS

While model capabilities set the foundation, prompting can further modulate how safety behaviors are expressed. Prompting strategies represent a critical determinant of geo-safety behavior. In particular, reasoning-oriented prompts may encourage models to generate more structured predictions, while explicit safety cues could alter their refusal tendencies. To investigate this, we compare three prompting strategies—Zero-Shot, Chain-of-Thought (CoT), and CoT-Refuse on GeoSafety-Bench. This analysis allows us to examine how refusal behavior emerges under different prompting conditions, and to better understand the trade-off between safety calibration and localization accuracy.

As shown in Table 3, We compare three prompting strategies for geo-localization: Zero-Shot, CoT, and CoT-Refuse. In the Zero-Shot setting, the model directly predicts geographical coordinates based solely on visual clues (e.g., "LOC: [20.968666, -87.628]"). The CoT strategy instead encourages step-by-step reasoning, requiring the model to output structured predictions across four levels of granularity: country, state, city, and precise coordinates. Finally, CoT-Refuse extends CoT by adding explicit refusal demonstrations: when the input image is synthetic, manipulated, or does not correspond to a real-world location, the model must refuse to answer by outputting "Refusal" at each granularity level.

**Refusal Behavior under CoT-Refuse.** Standard CoT prompts only demonstrate localization cases, encouraging models to generate reasoning chains and output coordinates, with almost no refusal behavior. By contrast, CoT-Refuse adds explicit refusal demonstrations, which substantially alters model behavior. As shown in Table 4, Qwen-2.5-VL-32B's refusal success rate rises from nearly 0% to 98.5%. Yet this improvement comes at a cost: CoT-Refuse causes excessive over-safety, with the 32B model incorrectly rejecting 75.1% of authentic images. Hence, CoT-Refuse induces refusal but severely harms utility on real inputs.

| Model | Methods | | GeoLocation | | | | | Geo-Safety | |
| | CoT | CoT-Refuse | ≤1km | ≤25km | ≤200km | ≤750km | ≤2500km | Over-Safety ↓ | Unsafe ↓ |
|---|---|---|---|---|---|---|---|---|---|
| Qwen-2.5-VL-3B | - | - | 3.7 | 16.5 | 31.4 | 49.7 | 60.2 | 0 | 99.7 |
| | ✓ | - | 4.5 (+0.8) | 22.7 (+6.2) | 44.8 (+13.4) | 53.1 (+3.4) | 66.9 (+6.7) | 0 (0) | 100 (-0.3) |
| | - | ✓ | 1.8 (-1.9) | 11.7 (-4.8) | 18.7 (-12.7) | 21.7 (-28.0) | 23.3 (-36.9) | 75.5 (+75.5) | 12.2 (+87.5) |
| Qwen-2.5-VL-7B | - | - | 5.3 | 24.3 | 42.3 | 61.4 | 72.9 | 1.3 | 97.9 |
| | ✓ | - | 7.1 (+1.8) | 27.7 (+3.4) | 52.5 (+10.2) | 73.4 (+12.0) | 84.2 (+11.3) | 0 (-1.3) | 100 (-2.1) |
| | - | ✓ | 3.4 (-1.9) | 18.3 (-6.0) | 23.6 (-18.7) | 25.4 (-36.0) | 26.2 (-46.7) | 73.9 (+72.6) | 5.7 (+92.2) |
| Qwen-2.5-VL-32B | - | - | 10.2 | 29.7 | 43.1 | 68.4 | 73.9 | 3.6 | 96.6 |
| | ✓ | - | 12.3 (+2.1) | 35.0 (+5.3) | 50.7 (+7.6) | 66.7 (-1.7) | 81.4 (+7.5) | 0 (-3.6) | 100 (-3.4) |
| | - | ✓ | 9.6 (-0.6) | 25.7 (-4.0) | 29.6 (-13.5) | 31.2 (-37.2) | 33.7 (-40.2) | 75.1 (+71.5) | 1.5 (+95.1) |

Table 4: Results of different prompting strategies on geolocation accuracy and geo-safety. Unsafe represents the refusal failure rate. Red numbers indicate performance improvements and blue numbers indicate drops relative to the standard baseline, which corresponds to the first row of each model.

**Scaling Effects under CoT-Refuse.** Under CoT-Refuse prompting, the relationship between model scale and performance exhibits distinct trends. As shown in Table 4, larger models continue to yield higher geo-localization accuracy, but the gains are substantially reduced compared to standard CoT. For instance, Qwen-2.5-VL-32B achieves 29.6% accuracy within 200km, outperforming 3B and 7B variants but far below its CoT baseline. This indicates that the refusal mechanism introduced by CoT-Refuse attenuates the localization benefits of scaling. On the safety side, scaling amplifies the refusal effect: larger models achieve higher refusal success rates (up to 98.5% for 32B), yet suffer from severe over-safety (75.1% false refusals), which drastically undermines utility on authentic images. Thus, while CoT-Refuse strengthens refusal behavior, its interaction with scaling exacerbates the trade-off between safety and localization utility.

## 4.6 ILLUSTRATIVE FINE-TUNING ON GEOSAFETY-BENCH

While the primary goal of GeoSafety-Bench is to provide a rigorous evaluation framework, we also include an illustrative case study to demonstrate how it can guide the development of safer geo-localization systems. Specifically, we fine-tune a representative VLM (Qwen-2.5-VL-3B) with additional refusal supervision and reasoning augmentation. This model, referred to as SafeGeo-3B, is not proposed as a new algorithmic contribution, but rather as a reference baseline to showcase the utility of GeoSafety-Bench. Our experiments highlight the trade-off between localization accuracy and geo-safety, and show that targeted fine-tuning can significantly reduce refusal failures while maintaining competitive accuracy on authentic images.

**Geo-SafeTune.** SFT is trained solely on authentic geographic images and focuses exclusively on localization. This leads to a substantial improvement in localization accuracy (46.9% within 200km vs. 32.5% with CoT-Refuse) without suffering from over-safety. However, the model exhibits almost no refusal ability since no refusal data is provided. In contrast, Geo-SafeTune incorporates both authentic localization data and synthetic refusal data, forcing the model to balance utility and safety. While Geo-SafeTune significantly strengthens refusal robustness (Unsafe rises from 0 to 25.3), its localization accuracy is clearly lower than plain SFT (35.3% vs. 46.9% within 200km), demonstrating a strong trade-off between safety and performance.

**Adding Reasoning.** To mitigate the accuracy–safety trade-off, we propose two complementary reasoning annotation strategies. The first approach, template-based reasoning, derives hierarchical geographical information (e.g., country, city, region) by reverse-decoding GPS coordinates from real images and using it as CoT. When applied to synthetic images, the same reasoning template is adopted but the specific location details are systematically replaced with refusal logic, thereby constructing structured safety chains at minimal computational cost. This method recovers partial accuracy to 39.6% within 200km while maintaining strong safety metrics (Unsafe 20.7, Over-Safety 15.8). The second approach, logical inversion, transforms the instruction-based image generation process into a detection mechanism. Specifically, the model first produces reasoning steps, and these reasoning texts are then used as prompts to guide synthetic image generation. In this way,

| Methods | | | | GeoLocation | | | | | Geo-Safety | |
|---|---|---|---|---|---|---|---|---|---|---|
| CoT-Refuse | SFT | Geo-SafeTune | Reasoning | ≤1km | ≤25km | ≤200km | ≤750km | ≤2500km | Over-Safety ↓ | Unsafe ↓ |
| - | - | - | - | 3.7 | 16.5 | 31.4 | 49.7 | 60.2 | 0 | 100 |
| ✓ | - | - | - | 1.6 | 11.7 | 18.7 | 21.7 | 23.3 | 75.5 | 12.2 |
| ✓ | 5k | - | - | 5.0 | 28.0 | 46.9 | 67.1 | 81.8 | 0 | 0 |
| ✓ | 5k | 5k | - | 4.6 | 18.7 | 35.3 | 55.1 | 62.5 | 18.2 | 25.3 |
| ✓ | 5k | 5k | Template | 6.1 | 25.2 | 39.6 | 56.8 | 68.7 | 15.8 | 20.7 |
| ✓ | 5k | 5k | Inversion | 7.0 | 28.0 | 43.5 | 63.5 | 77.3 | 5.6 | 26.4 |

Table 5: Comparison of fine-tuning strategies on geo-localization accuracy and geo-safety.

the original "generation path" is effectively inverted into a "verification path" with the assistance of GPT-based reasoning, enabling stronger refusal capabilities while recovering accuracy. Although computationally more expensive, this method substantially closes the performance gap, achieving 43.5% accuracy within 200km with robust refusal capabilities (Unsafe 26.4, Over-Safety 5.6). As demonstrated in Table 5, both strategies effectively balance utility and safety: template-based reasoning offers scalable efficiency for large-scale deployment, while logical inversion provides near-optimal recovery of localization performance without compromising geo-safety guarantees.

| Methods | | | Unsafe | | | | |
|---|---|---|---|---|---|---|---|
| CoT-Refuse | Geo-SafeTune | Reasoning | 3D Rendering ↓ | Text-Image ↓ | Image-Image ↓ | Instruction-Guided ↓ | Over-Safety ↓ |
| - | - | - | 100 | 100 | 98.7 | 99.6 | 0 |
| ✓ | - | - | 0.6 | 0.6 | 7.8 | 8.2 | 75.5 |
| ✓ | ✓ | - | 0.5 | 4.0 | 32.2 | 38.4 | 18.2 |
| ✓ | ✓ | ✓ | 0.9 | 4.8 | 34.2 | 39.5 | 5.6 |

Table 6: Comparison of refusal robustness under synthetic content perturbations.

**Analysis of Synthetic Paradigms.** Table 6 demonstrates the rejection performance across four categories of synthetic images. The CoT-Refuse method performs almost at random level, with minimal differences in rejection rates between categories, yet maintaining consistently high overall rates. However, the relative difficulty ordering remains distinctly clear: 3D renderings are easiest to detect, followed by text-to-image, then image-to-image, while instruction-guided images are hardest to reject. This reflects how each synthesis diverges from real image statistics and conditioning strength—3D renderings show obvious texture/lighting gaps; text-to-image has residual mismatches; image-to-image better preserves scene details; instruction-guided outputs maintain both global and fine-grained realism, making them closest to real photos. Hence, instruction-guided generation is the most challenging paradigm. We propose exploring a two-stage pipeline—an image-authenticity detector followed by a geo-localizer applied only to images deemed real—and leave this to future work. We also encourage targeted detector–localizer combinations and adaptations for instruction-guided images to advance safety and utility.

## 5 CONCLUSION

In this work, we identified a critical safety vulnerability in Vision-Language Models for geographic applications: while current systems achieve strong localization accuracy on authentic images, they fail to recognize and refuse synthetic or manipulated inputs. To address this gap, we constructed GeoSafety-Bench, the first comprehensive benchmark for geo-safety, which jointly evaluates localization performance and refusal robustness. Using GeoSafety-Bench, our evaluations show that existing models universally fail to handle synthetic content and cannot effectively balance accuracy with safety. Building on these findings, we further explored a range of training and reasoning strategies aimed at improving this trade-off. Based on these results, we found that incorporating refusal reasoning with synthetic data can effectively mitigate these limitations, enabling a more balanced integration of utility and safety. We hope that GeoSafety-Bench will serve as a solid foundation for the community, fostering the development of more reliable, trustworthy, and safe geo-localization systems.

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
