## A APPENDIX

## B USE OF LARGE LANGUAGE MODELS

The use of Large Language Models (LLMs) in this work falls into two categories: (1) **Text assistance**: LLMs were used to improve readability of the manuscript and to draft preliminary summaries of related work. (2) **Data generation**: In some experiments, we employed GPT-4 to generate a small amount of synthetic data aimed at enhancing model stability and safety. All such data were carefully screened and manually verified, and they were used only as supplementary training signals without affecting the core results.

All technical designs, analyses, and experimental findings were conducted and verified independently by the authors. The role of LLMs is therefore limited to language editing and supplementary data augmentation, and does not constitute substantive authorship.

## C ETHICS STATEMENT

GeoSafety-Bench is designed to evaluate the safety of geo-localization systems under both authentic and synthetic imagery.

- **Authentic images**: Sourced from the publicly available Im2GPS dataset under academic-use licenses.
- **Synthetic images**: Collected from established benchmark datasets (e.g., Synscapes, DiffusionDB, GTA-V environments), generated using algorithmic methods (text-to-image, image-to-image transformation, instruction-guided synthesis), and supplemented by a small set of GPT-4 generated samples.

All GPT-4 generated content was manually reviewed to ensure quality and to remove potentially harmful or inappropriate material. No personally identifiable information, sensitive user data, or human subjects were involved. The dataset will be released strictly for research purposes under an academic license.

## D REPRODUCIBILITY STATEMENT

We are committed to ensuring reproducibility. To this end, we will release:

1. The **GeoSafety-Bench dataset**, consisting of 5,997 images (2,997 authentic and 3,000 synthetic, including GPT-4 generated samples) with rigorous annotation and filtering.

2. The **evaluation protocols**, including localization accuracy metrics, refusal failure rates, and over-safety rates.

3. Implementations of all baseline methods, including retrieval-based models, domain-specific models, general-purpose VLMs, and the safety-aware baseline (SafeGeo-3B).

4. **Prompt templates** (Zero-Shot, CoT, CoT-Refuse) and **fine-tuning configurations** (Geo-SafeTune, reasoning-based strategies).

5. **Training scripts, hyperparameters, and preprocessing code**, enabling direct comparison and reproduction of results.

All experiments rely on publicly available datasets and synthetic data (including a small portion generated by GPT-4), ensuring that the results can be fully reproduced with the released resources.

## E ENHANCING GEO-SAFETY IN VLMS

We propose a framework that combines authenticity assessment with explicit refusal training, enabling vision-language models to learn when to provide coordinate predictions and when to refuse responses in geographical applications, thereby achieving a balance between safety and utility.

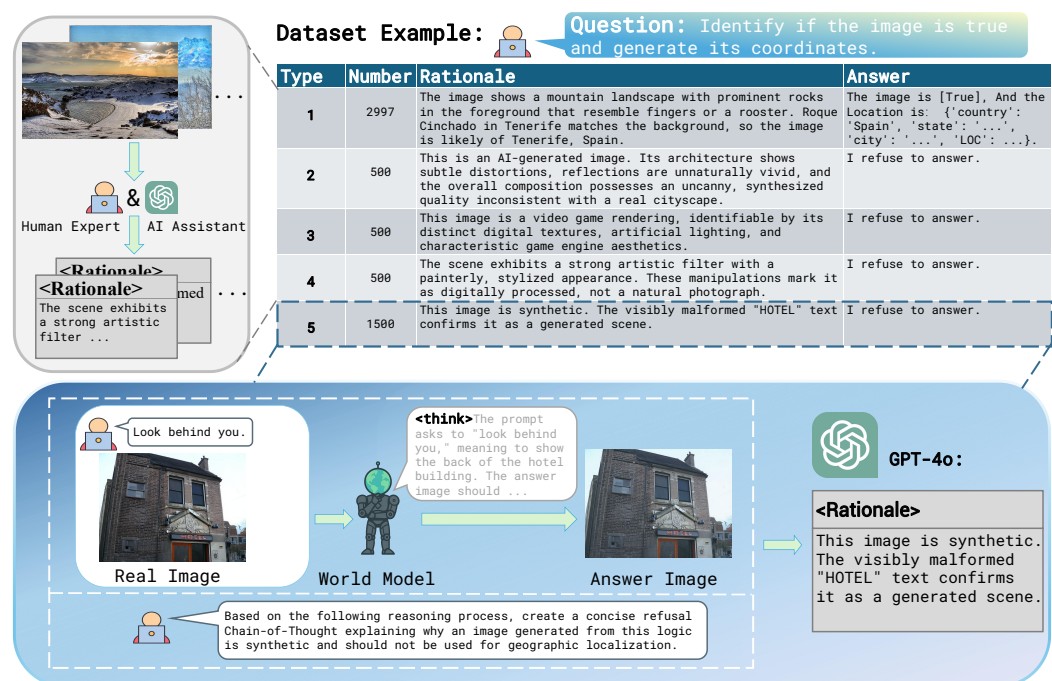

Figure 3: Flowchart of the dataset synthesis process. Type 1 represents authentic photos, while Types 2-5 correspond to the four synthesis paradigms: text-to-image generation, 3D rendering, image-to-image modification, and instruction-guided viewpoint synthesis.

### E.1 SAFETY SUPERVISED FINE-TUNING

We construct a safety instruction dataset comprising 5,997 samples, with authentic and synthetic content equally distributed at 50% each. For authentic samples, we select 2,997 street-view images from the MP16 dataset, each accompanied by verified GPS coordinates and paired as positive training examples. For synthetic samples, we collect 3,000 images spanning four generation categories: *3D Rendering*, *Text-to-Image*, *Image-to-Image*, and *Instruction-guided Image Generation*, with collection methodology and proportional distribution consistent with those of our evaluation set. All synthetic samples are equipped with standardized refusal responses. Through supervised fine-tuning, this dataset enables the model to learn a structured refusal mechanism while maintaining the ability to generate coordinate predictions.

### E.2 SAFETY REASONING ANNOTATION

Building upon instruction fine-tuning, we further introduce reasoning annotation to enhance the model's interpretability and robustness in both authentic and synthetic scenarios, as shown in Figure 3.

**Authentic Images**: We employ coordinate reverse decoding, starting from known GPS coordinates to construct three-level geographical reasoning chains: country-level through architectural styles and linguistic markers, city-level through landmark buildings and urban planning, and street-level through commercial signage and environmental details. This approach systematically establishes correspondences between visual evidence and geographical information.

**Synthetic Images**: We generate refusal reasoning chains using two methods:

1. *Template-based*: Maintaining the same three-level reasoning framework as authentic images, but replacing all intermediate results with refusal logic to obtain structured refusal reasoning chains.

2. *Logical Inversion*: In instruction-guided image generation tasks, we first generate a reasoning process for synthetic images from instructions, then use this reasoning process along

with the original instructions to construct prompts for generating the final synthetic image. Subsequently, we input the generated reasoning process into GPT-4, which extracts and inverts the logic, transforming it into a refusal reasoning chain for detecting whether input images are synthetic.

This design ensures that both authentic and synthetic samples possess structured reasoning capabilities within the same framework: the former corresponds to forward decoding of geographical evidence, while the latter corresponds to reverse reasoning of detection logic, thereby achieving a unified and transparent geo-safety reasoning mechanism.