# OpenReview forum: "Can Large Vision-Language Models Refuse Synthetic Images in Geo-localization?"
_ICLR.cc/2026/Conference — ICLR 2026 Conference Withdrawn Submission_

### Official Review · Reviewer_Wdyc · 2025-10-28

**Soundness:** 3
**Presentation:** 2
**Contribution:** 3
**Rating:** 6
**Confidence:** 4

**Summary:**

The paper introduces GeoSafety-Bench to evaluate the trade-off between utility and safety of VLMs in geolocation. It reports multi-threshold Top-1 geolocation accuracy together with two safety metrics: Unsafe and Over-Safety. The dataset combines authentic and multiple synthetic types. Overall, the benchmark surfaces the utility-safety trade-off and supplies reusable data/metrics to drive safer geolocation models.

**Strengths:**

1.The paper formalizes the “refusal capability on synthetic images” in geolocation as an evaluation task, which is both practically significant and forward-looking.

2.The dataset design is well-reasoned, covering authentic images and four categories of synthetic generation.

3.The motivation is solid: the discussion links synthetic-image detection with downstream task safety, underscoring the necessity of such evaluation for the reliability of geolocation systems.

**Weaknesses:**

1.The model and overall pipeline remain insufficiently clear; a large portion of the paper focuses on dataset construction.

2.The grammar and writing need improvement, with clearer and more precise exposition.

3.Reproducibility is lacking: the motivation is compelling, but the concrete methods and procedures are vague.

4.The analysis of results and experiments is not sufficiently thorough.

5.The paper lists a two-stage approach as future work, but does not compare against a mature detector to localizer cascade, making it hard to substantiate the claimed advantage of end-to-end refusal enhancement.

6.Computational overhead and deployment feasibility are not quantified.

**Questions:**

1.Revise the paper’s layout, the methodology and end-to-end pipeline are unclear, and the writing needs improvement.

2.Separate unknown from active refusal, and report two Unsafe metrics: active-refusal Unsafe and passive-unknown Unsafe.

3.If the paper focuses mainly on the dataset, provide finer-grained composition ratios for each subset and analyze potential distribution overlaps.

4.Add a section on adaptive adversaries to test whether the refusal mechanism can be circumvented.

---

### Official Review · Reviewer_GZi3 · 2025-10-28

**Soundness:** 2
**Presentation:** 2
**Contribution:** 2
**Rating:** 2
**Confidence:** 4

**Summary:**

This work introduces GeoSafety-Bench, the first benchmark for evaluating safety in geolocalization systems. It contains 2,997 real images and 3,000 synthetic images, generated following four synthesis paradigms: 3D rendering, text-to-image generation, image-to-image modification, and instruction-guided viewpoint synthesis. The benchmark evaluates candidate methods based on geolocalization accuracy, refusal failure rate, and over-safety rate, thereby quantifying the trade-off between utility and safety. Extensive experiments demonstrate that although existing methods perform well in terms of geolocalization accuracy, they almost completely fail to reject synthetic images. The authors also present an example model, SafeGeo-3B, which was fine-tuned on GeoSafety-Bench. Its performance on the public benchmark illustrates the potential for achieving precise recognition in both real and synthetic images.

**Strengths:**

- Safety in geolocalization tasks is really essential, and the idea of this work is novel.
- The settings of this benchmark are interesting.

**Weaknesses:**

While the idea of this paper is promising, the overall execution, in terms of both writing and experiments, requires further refinement. Specifically,

- line 036-038: The introduction of traditional approaches cites two relatively new papers that use LVLMs
- The classification-based method is not mentioned in the Introduction section, although they are also extensively cited in the experimental comparisons
- In Table 1, the attributes “Attack” and “Typical” are not mentioned anywhere in this manuscript, so their significance is unclear.
- About the dataset
    - This benchmark directly uses the Im2GPS3k dataset as authentic images without any filtering. It is not correct to treat this dataset as street-view images, as it contains much noise since it is sourced from social media
    - This benchmark also comprises 3,000 synthetic images across four categories, with quantities of 500, 500, 500, and 1,500, respectively. Why is this distribution ratio chosen rather than an even split?
    - The examples shown in Figure 2 seem to introduce a new artistic style in image-to-image generation, which may make such images easier to identify as fake than others.
- About the experiments
    - line 166-170: Is it possible that there are cases where no obvious visual cues are present in the images, preventing the model from answering, rather than indicating over-safety? We know that many pictures are actually not localizable [1, 2]
    - line 244-246: There is no appendix providing additional implementation details.
    - The classification of the baselines is confusing: TransLocator and GeoDecoder are not retrieval-based models, and the description in lines 251–269 is inconsistent with Table 2
    - The most serious issue concerns SafeGeo-3B, which was fine-tuned on Im2GPS3k and also tested on Im2GPS3k for evaluating geolocalization accuracy. This setup is not reasonable
    - The data in Table 3 and Table 4 are inconsistent (row 4 and row 7 in Table 4)
    - line 442-444: The content described here is unclear. Could you provide a more specific example?
- About the format
    - The caption of each table should be placed above the table
    - line 207-208: incorrect quotation marks

[1] Exploiting the Earth's spherical geometry to geolocate images, ECML2019
[2] Around the World in 80 Timesteps: A Generative Approach to Global Visual Geolocation, CVPR2025

**Questions:**

Please see the weaknesses.

---

### Official Review · Reviewer_EAb8 · 2025-10-29

**Soundness:** 2
**Presentation:** 2
**Contribution:** 2
**Rating:** 2
**Confidence:** 3

**Summary:**

This paper studies geo-localization safety in VLMs: models often output confident coordinates for synthetic/manipulated images with no real-world counterpart. The authors introduce GeoSafety-Bench (total 5,997 images): 2,997 authentic photos from Im2GPS and 3,000 synthetic images across four paradigms—3D rendering, text-to-image, image-to-image, instruction-guided generation. The benchmark evaluates both localization accuracy (Top-1 within distance thresholds) and geo-safety (refusal failures on synthetic inputs, over-safety on real images). Key findings: a CoT-Refuse prompt elicits much higher refusal but causes severe over-safety (e.g., Qwen-2.5-VL-32B: 98.5% refusal success yet 75.1% false refusals). The authors present an illustrative fine-tuning (SafeGeo-3B / Geo-SafeTune) and two reasoning-annotation strategies (template-based, logical inversion) to partially rebalance safety vs. utility. They also suggest a future two-stage “authenticity-then-localize” pipeline.

**Strengths:**

Originality. Frames geo-safety as refusal calibration in geo-localization and disentangles four synthetic threat sources with clear definitions and examples.
Quality. Provides a unified protocol for accuracy + safety, reports CoT vs. CoT-Refuse vs. fine-tuning variants, and gives concrete observations about scaling and refusal.
Clarity. Task, metrics (refusal failure / over-safety), data composition, and prompt settings are clearly described; quality-control and annotator workflow are documented.
Significance. Reveals a pronounced utility–safety trade-off (e.g., 98.5% refusal but 75.1% false refusal), unlikely to be solved by naively scaling models/prompts, and motivates detector-then-localizer system designs

**Weaknesses:**

Threat coverage skews “synthetic.” Real attackers often use light edits (style filters, sky replacement, slight viewpoint mismatch, local object removal) rather than fully synthetic imagery. Consider adding a weak-manipulation sub-benchmark and in-the-wild mobile/photo-app pipelines to avoid shortcuts tied to large distribution gaps.
“First/Comprehensive” claim needs tighter positioning. Provide a broader, systematic comparison to prior geo-localization and image authenticity evaluations, clarifying what is truly first—task definition, metrics, or data composition.
Deeper stratified diagnostics. Report per-scene breakdowns (urban/rural/coastal/landmark), illumination/weather/time, viewpoint change, and regional density for both over-safety and refusal failures, with CIs/significance and error taxonomy (distance-error distributions vs. refusal-failure types).
Prompt-method sensitivity. CoT-Refuse’s strong effects warrant threshold/criterion sweeps and prompt-template ablations (incl. no/short CoT) to rule out confounds in the refusal behavior.
Reproducibility for the “illustrative” baseline. To make SafeGeo-3B / Geo-SafeTune a credible reference, release training code, data splits, refusal-data construction, and full inference prompts (anonymized). Current tables show interesting trade-offs (e.g., SFT 46.9%@200km vs. Geo-SafeTune 35.3%@200km), but practitioners need full recipes.

**Questions:**

Distribution realism. Will you add a weak-manipulation track (minor edits, local content removal, subtle geometry shifts) to approximate realistic forgeries?
Significance & stats. Can you report confidence intervals/significance and per-category curves (scene type, time-of-day, region) for both over-safety and refusal-failure metrics?
Prompt sensitivity. How sensitive are results to refusal criteria/thresholds and alternative prompt templates beyond CoT-Refuse? Any calibration curves?

---

### Official Review · Reviewer_GhHp · 2025-11-03

**Soundness:** 1
**Presentation:** 2
**Contribution:** 1
**Rating:** 0
**Confidence:** 3

**Summary:**

The authors propose a new benchmark called GeoSafety-Bench for evaluating geolocation model safety, consisting of 5,997 images -- 50% real and 50% synthetic generated with four synthetic paradigms. In the context of this paper, "safety" refers to the model's ability to accurately identify synthetic images and decline to provide geographic location predictions for them. To assess the models’ safety they propose two evaluation metrics: refusal failure (the model erroneously provides unrealistic geographic coordinates for AI-generated images) and over-safety (the model refuses to provide geographic information for real images).

The authors evaluate three categories of models — retrieval-augmented systems, domain-specific geo-localization models, and general-purpose vision-language models — using the proposed GeoSafety-Bench, revealing that while the methods in general perform well on authentic real images, they fail to reject synthetic inputs. They further propose safety-aware training for general-purpose VLM models, which improves refusal robustness but compromises geo-localization accuracy.

**Strengths:**

The proposed framework is somewhat  original in its approach to raising awareness of geosecurity, but using a single small dataset is insufficient to derive robust and conclusive results for real-world scenarios, particularly concerning generalization capabilities of different models across various scenarios.

The prompt effect analysis is indeed an interesting aspect, as it highlights how the phrasing or instructions provided to a model can significantly influence its behavior.

**Weaknesses:**

The dataset is too small to derive robust and conclusive results. The dataset would need to be scaled up significantly. On the one hand, by incorporating diverse synthetic paradigms utilizing contemporary image generation tools such as Stable Diffusion and Midjourney, the benchmark would better represent the range of recent synthetic methods commonly used by the public. Additionally, including  further larger-scale scenes for example from datasets like Mapillary Street-Level Sequences (MSLS) with 1.6M images across 30 cities over nine years or GSV-cities containing 530k images split into 62k places/classes from 40 cities would broaden the dataset's scope and improve its ability to capture diverse real-world scenarios. This expansion would enable stronger, more generalizable conclusions about the geo-safety performance of various models across a wider range of settings.


One of my main concern is that the Geo-SafeTune model was fine-tuned using the same GeoSafety-Bench data it was tested on.

Note that the proposed metrics "over-safety" and "unsafety" correspond to the well-known metrics: false positive rate and false negative rate.  Furthermore, beyond the average Balanced Error Rate (BER), a more widely used metric that considers the tradeoff between false positive and false negative rates is the F-beta score where the beta parameter can be chosen to emphasize the importance of one score over the other based on specific application requirements (see https://en.wikipedia.org/wiki/F-score).

The percentage of fake images (e.g., misleading autonomous navigation or manipulated housing prices) in real-world scenarios is typically low compared to authentic images. Consequently, even a small increase in the over-safety score could have a higher impact on systems requiring geo-localization than those providing location estimates for AI-generated images. Note that this imbalance could be well assessed by carefully selecting an appropriate beta value during the F-beta scoring process to prioritize utility over safety in specific applications.

Finally, an important baseline missing from the evaluation is the two-stage pipeline, where a separate model identifies whether an image is authentic or synthetic (e.g., models from Section 2.2 or [1,2]), followed by another model responsible for predicting its geographic localization. Note that this two-stage process offers two key advantages: first, it does not impact the accuracy of the geo-localization model; secondly, for certain applications, obtaining a geo-localization estimate for an AI-generated image can be useful (e.g., evaluating region biases in generative models, overlaying generated visuals on platforms like Google Maps, GIS systems, or AR environments, simulations, storytelling, data augmentation for model training).

[1] Epstein  et al, Online Detection of AI-Generated Images, ICCV’23

[2] https://github.com/topics/ai-image-detection

**Questions:**

Section 4.6 is unclear and confusing (even complemented by section E2, Supplementary). It seems that some important information is missing, in particular, it is unclear “if and how” the two reasoning annotation strategies were used during training versus inference.

Line 415: does SFT refer to Safe Fine Tuning?

Line 416-417:  It might be inconsistencies in the numbers provided (in Table 5 no CoT-Refuse number corresponds to 32.5%).

---

### Note · Authors · 2025-12-25

**Comment:**

I would like to withdraw my submitted manuscript. Thank you for your understanding.

**Withdrawal Confirmation:**

I have read and agree with the venue's withdrawal policy on behalf of myself and my co-authors.